# Kinematic Behavior of an Untethered, Small-Scale Hydrogel-Based Soft Robot in Response to Magneto-Thermal Stimuli

**DOI:** 10.3390/biomimetics8040379

**Published:** 2023-08-19

**Authors:** Wenlong Pan, Chongyi Gao, Chen Zhu, Yabing Yang, Lin Xu

**Affiliations:** 1Institute of Intelligent Flexible Mechatronics, Jiangsu University, Zhenjiang 212013, China; 2222103003@stmail.ujs.edu.cn (W.P.);; 2State Key Laboratory of Solid Lubrication, Lanzhou Institute of Chemical Physics, Chinese Academy of Sciences, Lanzhou 730000, China

**Keywords:** magnetic, temperature-sensitive hydrogel, untethered soft robot, bionic gastropod, gradient magnetic field

## Abstract

Fruit fly larvae, which exist widely in nature, achieve peristaltic motion via the contraction and elongation of their bodies and the asymmetric friction generated by the front and rear parts of their bodies when they are in contact with the ground. Herein, we report the development of an untethered, magnetic, temperature-sensitive hydrogel-based soft robot that mimics the asymmetric micro-patterns of fruit-fly-larvae gastropods and utilizes cyclic deformation to achieve directional peristaltic locomotion. Due to Néel relaxation losses of nanomagnetic Fe_3_O_4_ particles, the hydrogel-based soft robot is capable of converting changes in external alternating magnetic stimuli into contracting and expanding deformation responses which can be remotely controlled via a high-frequency alternating magnetic field (AMF) to realize periodic actuation. Furthermore, the Fe_3_O_4_ particles included in the hydrogel-based soft robot cause it to follow a gradient magnetic field in confined liquid environments and can be coupled with AMFs for the targeted release of water-soluble drugs or targeted magnetic hyperthermia therapy (MHT). We believe that such a controlled motion can enable highly targeted drug delivery, as well as vascular disease detection and thrombus removal tasks, without the use of invasive procedures.

## 1. Introduction

After billions of years of evolution via the survival of the fittest, nature has broadly nurtured worms, earthworms, ruler worms, sandworms, caterpillars, and other invertebrates to acquire hydrostatic skeletal fluid–solid coupled micromotion behavior via body contraction and elongation deformations. Through the contraction and elongation of their bodies, invertebrates such as geometrid worms, sandworms, and caterpillars have acquired a large hydrostatic skeletal fluid–solid coupled deformation motion behavior in the form of fluctuations. Compared with wheeled, tracked, and footed robots, bionic creeping robots have better motion stability, flexibility, and environmental adaptability [1,2,3]. In particular, bionic, peristaltic soft robots [4], which mimic the shape, structure, or movement patterns of natural mollusks, are made of soft materials that can withstand large strains, have multiple degrees of freedom and continuous deformation capabilities, and can change their shape and size arbitrarily over a wide range to adapt to changing and complex terrain contours, thus achieving efficient peristaltic movement in unstructured environments. Therefore, the bionic creeping soft robot has broad application prospects and unparalleled advantages in exploration and rescue, military reconnaissance, pipeline inspection, minimally invasive surgery, and other fields, especially in non-structured environments with complex structures and narrow lengths.

Bionic peristaltic soft robots, which are still in their infancy, are developing rapidly, finding new applications in the fields of materials, design, preparation, and control, but they still face many difficulties and challenges. The flexible materials and bionic structures applied to soft robots are very different from those of traditional robots, making traditional drive methods such as motors no longer applicable to soft robots. To meet the performance requirements of soft robots with respect to large deformations and high degrees of flexibility, most drive methods for soft robots use flexible materials that can withstand large strains as the drive source or drive medium, mainly gas drives [5,6,7], shape memory alloy (SMA) drives [8,9,10], and intelligent hydrogel drives [11,12,13], etc., depending on the drive mechanism.

However, the deformation of hydrogel actuators with a uniform structure under uniform external energy field stimulation is limited to simple isotropic contraction and expansion, which makes it difficult to mimic the complex deformation behavior of mollusks. Therefore, the application of non-uniform external energy field stimulation and the development of anisotropic (e.g., bilayer, gradient, and pattern-structured) hydrogel actuators have become research directions of extreme interest in academia [14,15]. For example, hydrogels can be used in controlled-drug-release applications [16,17]. At Dublin City University, Florea et al. modulated the walking behavior of a front leg dragging a hind leg on a guide rail with a ratchet structure via the reversible dissolution and contraction deformation of a hydrogel-based robot which was caused by the bright extinction of light [18], but the hydrogel began the contraction deformation after 5 min of irradiation with the white light, which is a long drive time. The researchers introduced magnetic nanoparticles into the temperature-sensitive hydrogel and used the magnetic nanoparticles, under the action of an alternating magnetic field due to the Néel relaxation mechanism and the production of heat, to increase the local temperature of the temperature-sensitive hydrogel, thus obtaining a magnetic, temperature-sensitive hydrogel. The volume phase change of the magnetic temperature-sensitive hydrogel can be controlled by adjusting the temperature of the hydrogel through an alternating magnetic field, thus producing deformation behaviors such as contraction and elongation [19,20,21]. Moreover, magnetic hydrogels are widely used in the biomedical field due to their good biocompatibility and wettability [22,23,24,25].

Although the above “arch-shaped” hydrogel-based soft robots have shown locomotor behaviors similar to invertebrates such as maggots and earthworms, overall, the locomotor behaviors were achieved via local stimulation or by constraining the temperature-sensitive hydrogel-based soft robot’s deformation via the ratchet structure of an external guide, and the locomotor speeds were governed by the diffusion rate of water molecules within the temperature-sensitive hydrogel networks.

Therefore, this study developed a magnetic, thermo-sensitive hydrogel-based soft robot with an ontogenetic gastropod structure by using the advantages of a magnetic field drive without an energy storage device, a remote control, and a highly effective output. The volumetric phase change behavior of the magnetic thermo-sensitive hydrogel is regulated by using a high-frequency alternating magnetic field as a medium for remote temperature control; it then realizes a ratchet structure without an external guide constraint, causing the soft robot to exhibit a controlled bionic creeping motion on smooth surfaces. At the same time, the hydrogel-based soft robot, which is embedded with soft magnetic Fe_3_O_4_ particles, can also follow the magnetic field gradient to move in a narrow liquid environment. In summary, these capabilities not only potentially improve the application of magnetic, temperature-sensitive hydrogel-based soft robots in biomedicine, such as in thrombus removal tasks, drug release, and targeted magnetic hyperthermia therapy (MHT), but they also provide a theoretical and experimental basis for the development of new millimeter-scale or even submillimeter-scale magnetic, temperature-sensitive hydrogel-based soft robots and remote drive control.

## 2. Materials and Methods

### 2.1. Preparation of the Magnetic, Temperature-Sensitive Hydrogel

A total of 1.13 g of N-isopropyl acrylamide (NIPAM, from Aladdin Biochemical Technology Co., Ltd., Shanghai, China) monomer and 0.03 g of N,N′-methylene bisacrylamide (BIS, from Aladdin Biochemical Technology Co., Ltd., Shanghai, China) crosslinkers were dissolved in 8 g of deionized water. A total of 0.3 g of lithium magnesium silicate nanoparticles (Laponite XLG, from BYK Additives Co., Ltd., Shanghai, China) was added to 2 g of de-ionized water. Fe_3_O_4_ was prepared in a Tween 20 (TWEEN^®^ 20, from Aladdin Biochemical Technology Co., Ltd., Shanghai, China) solution. The mixture was stirred for 10 min to obtain a homogeneous dispersion of the magnetite nanoparticles. The radical initiator (0.019 g of ammonium persulfate (APS, from Aladdin Biochemical Technology Co., Ltd., Shanghai, China)) and accelerator (60 μL N,N,N,N-tetramethyl ethylenediamine (TEMED, from Aladdin Biochemical Technology Co., Ltd., Shanghai, China)) were then added and mixed to initiate the redox reaction. After mixing, the solution was injected into a mold and allowed to stand overnight at room temperature to obtain magnetic, temperature-sensitive hydrogels. The same method was used to prepare magnetic hydrogels with 0.45 and 0.6 g of magnetic powder; in these samples, only the ferric oxide content was changed.

### 2.2. Hydrogel-Based Soft Robot Design

Three kinds of hydrogels with different magnetic nanoparticle contents (0.3, 0.45, and 0.6 g) were prepared separately, and the prepared hydrogels were poured into cylindrical molds 6 mm in diameter and 1 mm in depth; three samples with these dimensions were used to test the volumetric phase changes of hydrogels with different magnetic nanoparticles contents under magneto-thermal conditions. The hydrogels prepared with 0.45 g of magnetic nanoparticles were then poured into two rectangular molds, one with a length of 40 mm, a width of 10 mm, and a thickness of 1 mm, and a second mold with the same dimensions but with a gastropod inclination angle of 45°; both samples were left in the molds for 24 h. Finally, the hydrogels were removed from the molds and cut into soft robotic parts without gastropods that were 10 mm in length, 5 mm in width, and 1 mm in thickness, and parts with gastropods that were 25 mm in length, 5 mm in width, and 1 mm in thickness and had a gastropod inclination angle of 45°.

### 2.3. Magnetic Application and Motion Platform

During testing, we placed the soft robot on a glass plate on which the vertical distance between the coil and the soft robot was about 10 mm, and the mandrel was perpendicular to the soft robot. The coil used here had 2 turns with a pitch of 6 mm, an inner diameter of 33 mm, and an outer diameter of 42 mm. The magnetic heating of the hydrogel was achieved by subjecting the hydrogel particles to an alternating magnetic field with a frequency of f = 417 kHz. In this paper, the magnetic field strength at the center of the coil was controlled at 20 kA m^−1^. Adjusting the induced current of the apparatus changes the strength of the magnetic field at the center of the coil. In the water-soluble-drug release experiment only, the magnetic field strength was 15 kA m^−1^ when the induced current was 15 A.

The motion platform for the gastropod-free hydrogel-based robot was prepared as follows: the A and B component gels of Ecoflex-10 silicone rubber were mixed at a ratio of 1:1, stirred well, poured into a rectangular mold with dimensions of 40 × 10 × 1 mm^3^ and an inclination angle of 45°, and left to stand for 4 h; the resulting structure was then removed from the mold. The hydrogel-based soft robot with gastropods was built on a glass plate with a length of 100 mm, a width of 100 mm, and a thickness of 5 mm.

## 3. Results and Discussion

### 3.1. Magnetic, Temperature-Sensitive Hydrogel Gel Mechanism

In this study, the magnetic, temperature-sensitive hydrogel-based soft robot was required to have a dual cross-linked hydrogel polymer network that is responsive to both magnetic fields and temperature variations; this was achieved via the preparation of a hydrogel composed of a cross-linked NIPAM network and Fe_3_O_4_ magnetic nanoparticles.

The temperature-sensitive polymeric monomer NIPAM reacted chemically with BIS, forming chemical bonds (primarily covalent bonds) and a chemically cross-linked network with stronger connections; lithium magnesium silicate nanoparticles (Laponite XLG), which have multifunctional lamellar structures themselves, generated large clusters during the gelation process (see Figure 1), as well as intermolecular forces. The physically cross-linked network formed is much weaker than a chemically cross-linked network, but most materials have a stable physical cross-linked network. Physical nanoclay networks exhibit strong viscoplastic behavior, and we found that the introduction of a nanoclay improves the strength and toughness of dual-network materials [26]. By utilizing these materials, a hydrogel was created which uses BIS and lithium magnesium silicate nanoparticles as cross-linking agents, with both chemically and physically cross-linked networks; this hydrogel is referred to herein as a magnetic, temperature-sensitive network (see Figure 1b).

As shown in Figure 1, when subjected to an alternating magnetic field, the magnetic, temperature-sensitive hydrogel-based soft robot containing magnetic nanoparticles will absorb the magnetic field’s energy due to Néel relaxation losses and convert this energy into heat; both the volume of the magnetic nanomagnetic particles within the hydrogel and the magnetic field strength will affect the hydrogel’s temperature change process in this relaxation process. The lower critical solution temperature (LCST) of NIPAM is around 32~35 °C. When the temperature is higher than the LCST, the magnetic, temperature-sensitive hydrogel transitions from a hydrophilic phase to a hydrophobic phase and produces a volume contraction. When the AMF is opened, the magnetic, temperature-sensitive hydrogel is heated above the LCST and the hydrogel undergoes a volume contraction, whereas when the AMF is closed, the hydrogel gradually cools down to below the LCST and absorbs water; this causes the hydrogel to return to its original volume. The effects of different magnetic nanoparticle contents on the volumetric phase transition of the hydrogel subjected to alternating magnetic fields were analyzed to determine the appropriate magnetic nanoparticle content for the preparation of magnetic hydrogel-based soft robots.

It was observed that an increase in the magnetic nanoparticle content had a significant effect on the temperature change of the hydrogel when the hydrogel was subjected to an alternating magnetic field (see Figure 2a). It was found that the diameter of the hydrogel containing 0.45 g of magnetic nanoparticles decreased by 1 mm after 10 min, the diameter of the hydrogel containing 0.3 g of magnetic nanoparticles decreased by almost 1 mm, and the hydrogel containing 0.6 g of magnetic nanoparticles only decreased by approximately 0.5 mm (Figure 2b). The highest magnetic nanoparticle content considered here (0.6 g) was found to have a higher heating rate and peak temperature compared with the other hydrogels (see Figure 2c). However, its volume phase change was smaller than that of the other two hydrogels. We believe that this is because too many magnetic nanoparticles encroach on the space of water molecules, and the reduction in its water content leads to the smallest volume change. Via infrared imaging, it was also observed that when subjected to an alternating magnetic field, the temperature change curve of the hydrogel was smoothest in the case of a magnetic nanoparticle content of 0.45 g (Figure 2d). The bulk phase change observed in this hydrogel was also considered beneficial in the creation of the hydrogel-based soft robots; in this work, we thus focused on the magnetic hydrogel with a magnetic nanoparticle content of 0.45 g.

### 3.2. Hydrogel-Based Soft Robots without a Gastropod Microstructure

The temperature change in the soft robot and the Ecoflex platform with a gastropod microstructure can be observed via infrared images: the temperature increased to 47 °C after 10 min, and the temperature varied approximately linearly with time in the time range of 1–9 min. The peak shown in Figure 3b is higher than the peak shown in Figure 2d is because the Ecoflex platform also contained 0.45 g of magnetic nanoparticles, and while the AMF heats the hydrogel-based soft robot, the Ecoflex platform is also heated (due to the material properties of the Eco-flex, no bulk phase changes were induced). The increase in the surface temperature with the volume phase change became more obvious, and the volume change of the hydrogel-based soft robot can be clearly distinguished in the infrared image shown in Appendix A. We then subjected the gastropod-free soft robot to cyclic experiments, i.e., repeated heating and cooling (see Figure 3c), on an Ecoflex platform with a gastropod microstructure to test its ability to travel unidirectionally on the surfaces of gastropod microstructures that generate asymmetric friction [8]. After 0.5 cycles, the gastropod-free hydrogel-based soft robot was found to have translated forward by a distance equal to 12.1% of its body length; after the AMF was turned off (keeping the soft robot in a wetted state), and after waiting for the system to reach a dissolution equilibrium state after the first cycle, it was found that the soft robot translated forward by a distance equal to 18.1% of its body length. We noted that the soft robot did not fully return to its initial state due to the diffusion rate of the water molecules within the temperature-sensitive hydrogel network. The second cycle was then performed, and after half a cycle of heating, the soft robot reached a peak forward translation equal to 21.2% of its body length, and the procedure followed in the first cycle was repeated; after the soft robot reached the dissolution and expansion equilibrium state, the soft robot without the gastropod hydrogel had translated forward by total of 9.7% of its body length across the two cycles.

### 3.3. Hydrogel-Based Soft Robots with a Gastropod Microstructure

To remove the ratchet structure of the external guide of the magnetic, temperature-sensitive hydrogel-based robot’s deformation, we placed a soft robot with a gastropod microstructure under the AMF and made it walk on a glass plate over two cycles. The foot microstructure of the soft robot was realized by mimicking the asymmetric micropatterns of fruit fly larvae [8,19]. We calibrated the positions of the front and rear feet of the soft robot with purple and blue dots (see Figure 4a). It can be observed that after one heating half-cycle, the soft robot exhibited a crumpled state, with the hindfoot buckling and crumpling forward by a distance equal to 17.04% of the body length of the robot; no significant change in the front foot was observed. After a total of one whole cycle, the soft robot absorbed water and returned to its equilibrium state following water absorption and swelling. After this complete cycle, the forefoot was seen to be displaced forward by a distance equal to 16.3% of the body length of the robot (Figure 4c), and the hindfoot was displaced forward by 5.3% (Figure 4d). We found that the head of the hindfoot sagged after the water absorption and swelling due to the difference in the structure of the ventral foot; thus, we used the forefoot, which had not changed significantly, as the benchmark for the measurements. After two complete cycles, it was found that as a whole, the soft robot had translated forward by a distance equal to 7.36% of its body length (Figure 4d).

Based on the experimental results, we found that although some displacements of the NIPAM hydrogel can be achieved by using the LCST of NIPAM in combination with the foot microstructure, these displacements were not obvious and required a long time.

### 3.4. Magnetic Navigation and Water-Soluble-Drug Release Experiments

Additionally, we demonstrated the ability of our magnetic, temperature-sensitive hydrogel-based soft robot to move rapidly in complex environments under a gradient magnetic field. A rubber hose was designed to simulate curved blood vessels in the human body (Figure 5a). The opening diameter of the rubber hose was a circle of 6 mm and the total length of the hose was 450 mm. A magnetic, temperature-sensitive hydrogel-based soft robot containing 0.45 g of magnetic powder was placed in the hose. Two irregular red stones of different sizes and weights, each representing a thrombus, were placed on the two paths of the hose. The weights of the thrombi in Path 1 and Path 2 were three and five times that of the soft robot. We used a cylindrical soft robot with a diameter of 3 mm for the thrombus removal experiments. During the experiment, the hose was placed on a 2 mm thick piece of cardboard, and the magnet was moved under the cardboard (Appendix A). As shown in Figure 5b, the soft robot reached the first bend in 3 s, pushed the small thrombus to the hose outlet in 6 s along Path 1, and then pushed the large thrombus to the hose outlet in 9 s along Path 2. As a result, the hydrogel-based soft robot can follow a gradient magnetic field to achieve rapid displacement and can push objects several times heavier than itself.

The soft robot demonstrated an excellent speed of 50 mm/s in the magnetic navigation experiment and successfully removed two thrombi. Furthermore, under an AMF, the robot can pass through a narrow hose (smaller than the robot’s initial size) after deswelling. Then, to verify the water-soluble-drug release ability of the soft robot, we combined the magnetic navigation experiment with a gradient magnetic field to move the soft robot to a specified position (Figure 6a) and demonstrated the water-soluble-drug release possibility of the soft robot by measuring the change in the weight of the hydrogel after opening the AMF.

We used 15 A-, 20 A-, and 28 A-induced currents for this test. The hose used in the test was 100 mm long and 6 mm in diameter, and the soft robot used in the test was 9 mm long, 3 mm wide, and 2 mm thick. As shown in Fig6. B and c, when the inductive current was set to 15 A, the temperature of the soft robot reached the LCST in about 6 min, and the temperature increased very slowly after that. When the inductive current was set to 20 A, the soft robot reached the LCST in about 3 min, and the maximum temperature reached about 34 °C. But, when the induced current was set to 28 A, the temperature of the soft robot increased very rapidly, exceeding the LCST in about 3 min and reaching a maximum temperature of nearly 40 °C. Therefore, boosting the induced current can significantly increase the warming rate and maximum temperature of the soft robot. The difference in the weight change of the soft robot between these three cases was also very obvious. As shown in Figure 6d, the weight of the soft robot decreased by about 45% after ten minutes when the induction current was set to 20 A, which is more than two times higher than the value achieved when the induction current was set to 15 A. Compared with 20 A, when the induced current is set to 28 A, the difference in the final weight change of the soft robot is only about 7%, but the release rate of the drug changes significantly. In addition, the size of the robot was reduced by about 30% at 28A-induced currents (Appendix A). In summary, we believe that the magnetic, temperature-sensitive hydrogel-based soft robot has the potential to release water-soluble drugs and pass through a narrow hose. And, with the enhancement of the induced current, the soft robot’s ability to release water-soluble drugs will become stronger.

## 4. Conclusions

In this paper, a small-scale magnetic, temperature-sensitive hydrogel-based soft robot with an ontogenetic bionic gastropod structure was developed; this structure exhibited the advantages of a magnetic field drive, including the absence of an energy storage device, a remote control, and a highly effective output. By using a high-frequency AMF as a remote temperature control stimulus, the temperature of the hydrogel is remotely controlled via the heat generated by Fe_3_O_4_ due to the Néel relaxation mechanism under the action of an alternating magnetic field which, in turn, regulates the volumetric phase-transition behavior of the magnetic thermo-sensitive hydrogel and realizes soft robots with controllable, biomimetic motions on smooth surfaces. Locomotion is enabled via two key mechanisms: (a) the periodic swelling and de-swelling of the hydrogels and (b) the asymmetric friction in the foot of the soft robots. The soft robot with a native bionic gastropod structure successfully removed the requirement of having a ratchet structure as an external guide for the motion of the magnetic, temperature-sensitive hydrogel-based soft robot. In addition, we demonstrated that the soft robot can move rapidly under a gradient magnetic field to remove thrombi (Figure 5) and release a water-soluble drug at fixed sites (Figure 6). Combined with the good biocompatibility of hydrogels, we believe that hydrogel-based soft robots have significant application potential for use in biology- and medicine-related fields. For example, they can be used as carriers for some water-soluble drugs and as external application tablets. Alternatively, through the miniaturization of these robots, combined with their fast magnetic navigation ability, they can be used to gain entry to the human body through blood vessels or the esophagus and use an AMF to achieve targeted drug release.

## Figures and Tables

**Figure 1 biomimetics-08-00379-f001:**
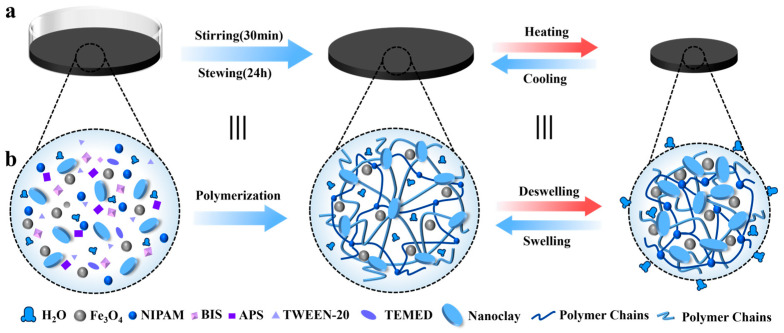
The preparation of the magnetic, temperature-sensitive hydrogel and a schematic diagram of the swelling/deswelling mechanism. (**a**) The preparation of the magnetic, temperature-sensitive hydrogel. (**b**) A schematic diagram of the internal structure network of the magnetic, temperature-sensitive hydrogel.

**Figure 2 biomimetics-08-00379-f002:**
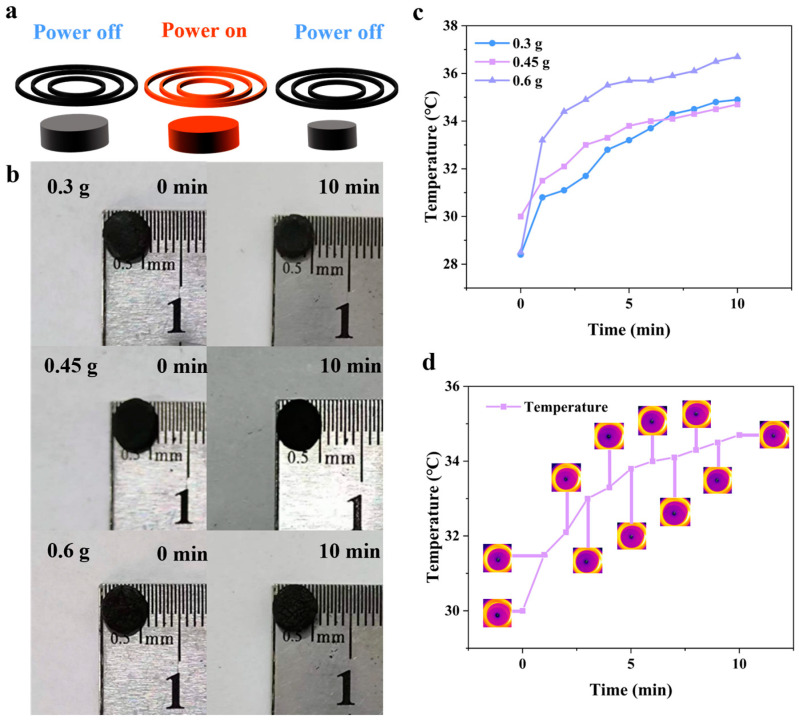
Temperature and volume changes of the Fe_3_O_4_-containing hydrogels with different compositions under a magneto-thermal coil. (**a**) Schematic of the hydrogel. (**b**) Volume changes of three Fe_3_O_4_ hydrogels. (**c**) Temperature dependence of the hydrogels. (**d**) Temperature variation of the hydrogel containing 0.45 g of magnetic nanoparticles.

**Figure 3 biomimetics-08-00379-f003:**
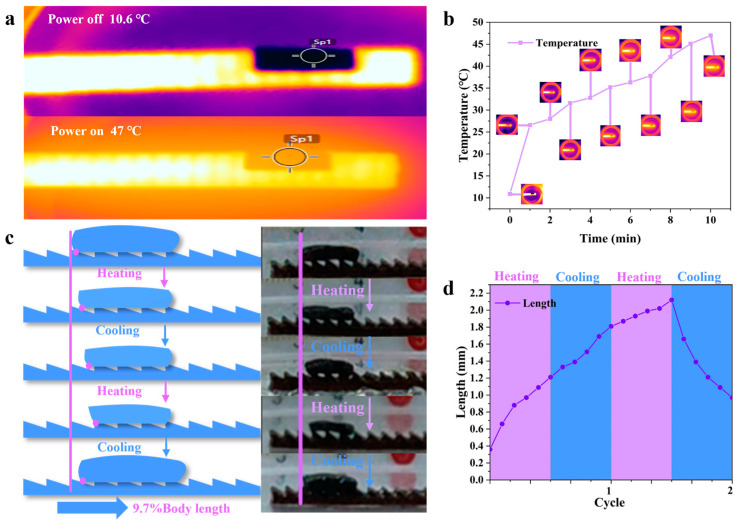
Movement of a gastropod-free soft robot on a gastropod microstructure under an AMF. (**a**) Infrared image of a gastropod-free soft robot walking on a gastropod microstructure. (**b**) Temperature profile. (**c**) Soft robot walking on gastropod microstructure. (**d**) The displacements of the soft robot.

**Figure 4 biomimetics-08-00379-f004:**
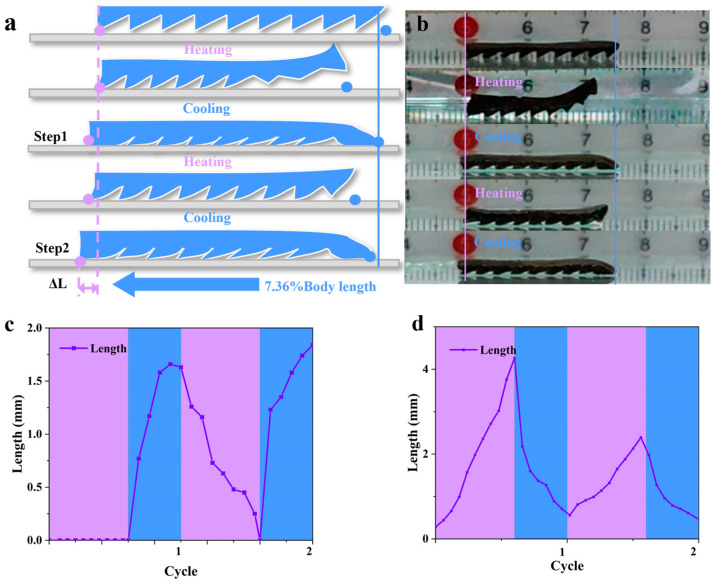
A soft robot with gastropods walking on a glass plate over two cycles under AMF. (**a**) Schematic of a soft robot with gastropods walking on a glass plate. (**b**) Soft robot with gastropods walking on a glass plate. (**c**) Forefoot displacement change; (**d**) hindfoot displacement change.

**Figure 5 biomimetics-08-00379-f005:**
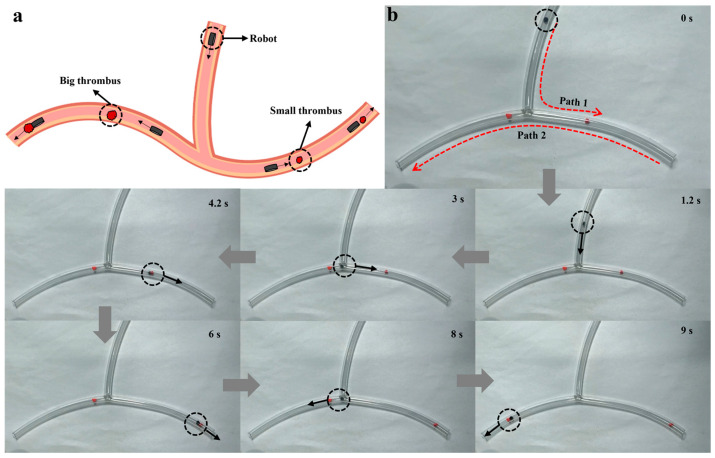
Use of a magnetic, temperature-sensitive hydrogel-based soft robot to simulate thrombus removal. (**a**) Schematic diagram of thrombus removal by the robot. (**b**) the process of thrombus removal by the robot (the arrows indicate the direction of the robot’s movement and the circles indicate the robot’s position).

**Figure 6 biomimetics-08-00379-f006:**
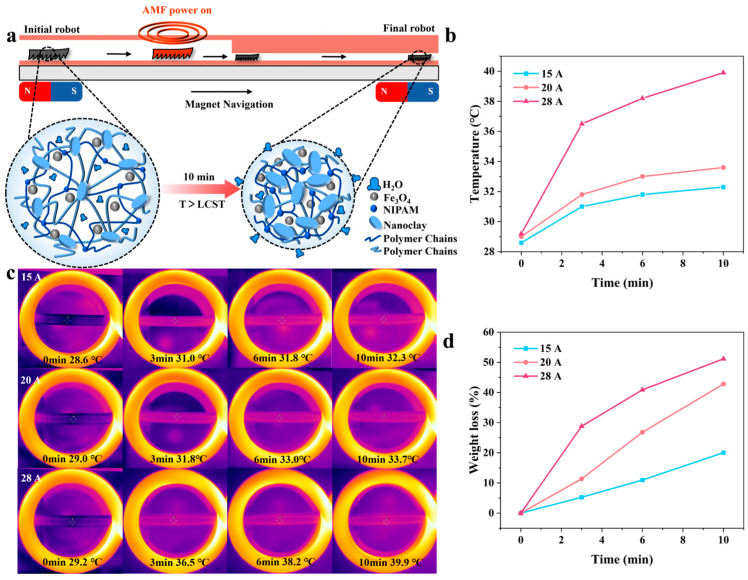
Magnetic navigation and water-soluble-drug-release experiments. (**a**) Schematic diagram of the drug-release principle of the soft robot under an AMF. (**b**) Temperature variation curves of the soft robot at induction currents of 15 A, 20 A, and 28 A. (**c**) Infrared image of a soft robot inside hose. (**d**) Weight reduction of the soft robot for induction currents of 15 A, 20 A, and 28.

## Data Availability

No new data were created or analyzed in this study. Data sharing is not applicable to this article.

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
