# Peer review of "Kinematic Behavior of an Untethered, Small-Scale Hydrogel-Based Soft Robot in Response to Magneto-Thermal Stimuli"

_biomimetics, 2023, doi:10.3390/biomimetics8040379_

Round 1

Reviewer 1 Report

In the manuscript, the kinematic behavior of an unbound dual-network hydrogel soft robot under magneto-thermal response was demonstrated. The synthesis of magnetic temperature-sensitive hydrogel, the design and magnetothermal actuation of the soft robot were described in detail. The manuscript is well written and organized. However, there are some items needed to be addressed before being considered for publication.

1. What the physical cross-linked network is in this dual-network hydrogel? Which materials were used?

2. In Fig. 1c and d, what the pink circles are?

3. In Fig. 3b and d, Fig. 4c and d, the font size should be identical with others and clear enough.

4. The paragraph 1 in “3. Results”, “This section may be divided by subheadings. It should provide a concise and precise 150 description of the experimental results, their interpretation, as well as the experimental 151 conclusions that can be drawn.” should be deleted.

5. “3. Results” should be “3. Results and discussion”.

6. The format of all references should be identical and at least according to the guidelines of the journal.

Reviewer 2 Report

The manuscript presents a hydrogel-based untethered robot that can be actuated remotely using alternating magnetic field. The actuating capability stems from the magneto-thermal effect of nanoparticles, Fe3O4, embedded in thermally shrinkable NIPAM structures. The Fe3O4 concentration was investigated to have an impact on actuating efficiency. And several demonstrations were performed to showcase the untethered locomotion and the movability in gradient magnetic field of their proposed soft robots.

First, the manuscript was not well written with some ambiguities and unclear statements. To name a few, P2L68, P3L114, P4L162, and P8L277. The language must be polished before considering it to be accepted. Followingly, the novelty is questionable. The demonstrations in Fig 3 and 4 are difficult to understand, and I did not see significant progress the authors made compared to other previously reported similar studies. Besides, no deep discussion was seen on the material characterizations and robotic design, and the manuscript lacks comprehensive studies, which make the entire structure quite weak. Overall, I do not suggest this manuscript to be accepted in its current form. It can be re-considered after a major revision. Please consider my following questions.

1. What is the function of the dual-network? I only saw the functionality of Fe3O4 and NIPAM, which can respectively offer magneto-heating and temperature-responsive deformation. But what is the meaning of having lithium magnesium silicate nanoparticles? Fig 1 cannot well distinguish the difference between the two networks.

2. The LCST of NIPAM is around 32~35C. It is a critical information and should be included in the content. Based on Fig 2c, the heat effect is too low to enable a dramatic volume change of NIPAM. It weakens the significance of this manuscript. Any thoughts on improving it?

3. The authors mentioned that both Fe3O4 concentration and alternative magnetic field strength can influence the temperature increase, but only studied Fe3O4 concentration. What were the experimental conditions of the alternative magnetic field? How close is the Magnetic thermal coil to the soft robot? What is the local magnetic field strength?

4. More clarification and quantification are needed in section 3.1, where Fig 2b is discussed. Why does the volume change of the 0.6g-case show the smallest?

5. In addition to the incomprehensible demonstrations in Fig 3c and 4a&b, the measured locomotions are too insignificant to convince me.

The manuscript was not well written with some ambiguities and unclear statements. To name a few, P2L68, P3L114, P4L162, and P8L277. The language must be polished before considering it to be accepted.

Reviewer 3 Report

There are too many writing errors in this article. Please make sure to use language editing services before the next submission.

Poor.

Author Response

Thank you for your valuable and thoughtful comments. We have also realized the lack of English writing, so we have used MDPI's English editing service. We have carefully checked and improved the English writing in the revised manuscript.

Round 2

Reviewer 2 Report

Thank you for addressing my questions. The manuscript is improved a lot, but I am still unsatisfied with the demonstrations regarding their showcase and significance. I hope the authors can think about better ways to demonstrate it in the next submission.

Reviewer 3 Report

1. Line 185 and 187. It seems other words are bold but Fe3O4 is normal.

2. Fig. 2(b). If you want to show the change in size (three dimensions) via figures, you should at least make these pictures look closer. I think the rule sizes are not the same for before and after figures, which makes it difficult to read. The authors are suggested to edit the figures. BTW, I think the size change isnt large enough.

3. Fig. 2. The font size difference between the figures is too large. Please try your best to make them the same. This kind of mistake is too unprofessional. Try to read more papers before your next submission. BTW, please also adjust the font size in Fig. 3 and Fig.4.

4. Line 232. The word Movement has a different font from other words.

5. I can't see the small pictures in Fig. 3(b) clearly. Please adjust the image size for easy reading.

6. The authors claimed they made robots as shown in Fig. 3 and Fig. 4. They should at least give us the velocities of their robots.

7. Line 20 and 21. The authors mentioned their robots can be used for biomedical applications, especially for endovascular interventions in the future. I think it's not a good idea to say something you can never realize in the abstract part. Besides, from the experimental results in Fig. 6, the final temperature is about 34 degrees centigrade, which is lower than our body temperature. Therefore, I think their material cannot be used for biomedical applications. The authors are suggested to remove this part or change your description.

The paper is easy to read now.
